# Position: The Age of AI Agents Demands A New Scientific Paradigm To Sustain Trustworthy Science

## Abstract

AI systems are becoming autonomous research agents that generate hypotheses, design experiments, and produce discoveries at scales beyond human oversight. As seen by increased submissions to ML venues, the verification gap between scientific output and our ability to check it is already widening, and autonomous agents make it worse by magnitudes given human-agent asymmetry. We argue that science must evolve its verification infrastructure, as it has before with peer review. However, while historical adaptations assumed human contributors who could be questioned and sanctioned, AI agents break this assumption. We propose criteria for an adapted verification infrastructure that emphasizes observable-by-default workflows, scalable verification, and clear attribution. We argue that without adaptation, ML and any scientific domain using agents face dangerous failures: experimental results that no person can verify, optimization for metrics over understanding, and accountability vacuums that erode scientific trust.

## 1. Introduction

AI systems are no longer just tools that scientists use—they are increasingly autonomous agents that generate hypotheses, design experiments, and produce discoveries at scales beyond human oversight. AlphaFold has predicted 200 million protein structures and won the 2024 Nobel Prize in Chemistry (Jumper et al., 2021). AlphaEvolve discovered the first improvement to matrix multiplication in 56 years (DeepMind, 2025). Sakana's AI Scientist generated complete research papers and passed peer review for a workshop in ICLR 2025 (Lu et al., 2024).

These systems share a defining feature: they reason autonomously, making thousands of decisions without human involvement. Our existing methods for verifying experiments do not account for such a system.

**We argue that the scientific method must evolve to address three interconnected challenges that AI agents create: observability (can we see what happened?), attribution (who is responsible?), and reproducibility (can we verify the results?).** These three mechanisms break down when research contributions happen alongside agents that scale rapidly and cannot be held accountable.

### 1.1. The Verification Gap Is Already Wide—Agents Will Make It Worse

The replication crisis revealed that scientific verification was already failing under human-speed science. Reproducibility is a defining feature of science, yet in 2015 it was found that only 36% of psychology studies could be replicated with statistically significant results (Open Science Collaboration, 2015). Nature found that across 1500 scientists, over 70% of researchers had failed to reproduce others' experiments (Baker, 2016). In machine learning, benchmark results often fail to replicate when hyperparameters are properly controlled (Pineau et al., 2021).

The verification gap is the distance between scientific output and our ability to verify it. Agents will make the gap dramatically worse.

METR's research shows that the length of tasks AI agents can complete with 50% reliability has doubled every seven months, accelerating to every four months in 2024-2025 (METR, 2025a). **If agentic capabilities continue to doubling every 4 to 7 months while human verification capacity remains constant, simple extrapolation suggests the gap could grow by a factor of $2^8$ to $2^{15}$ (roughly 250–30,000$\times$) within five years.** Even with significant slowdown in capability growth, the asymmetry is stark.

### 1.2. Three Challenges for Human-AI Science

Agent-enabled discovery creates three interconnected challenges. These extend the reproducibility crisis framework (Munafò et al., 2017) to account for contributors who cannot be meaningfully questioned or held accountable.

.**AUTHORERR: Missing \icmlcorrespondingauthor.**

*Proceedings of the 43rd International Conference on Machine Learning*, Seoul, South Korea. PMLR 306, 2026. Copyright 2026 by the author(s).

**Observability: Can we see what happened?** In control theory, a system is "observable" if its internal state can be inferred from outputs (Kalman, 1960). By this definition, AI agent reasoning is unobservable—we cannot reliably infer internal states. Agent observability can be improved via logs, metrics, and traces (Sridharan, 2018). However, current agent-assisted research lacks all three. We do not face this issue when working with other human researchers, because we can substitute direct observation with *social observation*: lab meetings, mentorship, questioning. With agents, not only are the internal states unreliable to infer, agents can also run for days or weeks autonomously, which can take too long for a scientist to review.

**Attribution: Who is responsible?** Agents cannot be held accountable. They have no careers, reputations, or incentives. For human science, *social accountability*—reputation, sanctions, career consequences—constrains even imperfect explanations. You can question a human collaborator and expect their answers to be constrained by consequences for being wrong. AI agents can be questioned, but their responses may not reflect actual reasoning processes (Huang et al., 2023; Bordt et al., 2025)—and there are no consequences for unreliable answers. We need technical attribution mechanisms.

**Reproducibility: Can we verify results?** The reproducibility crisis predates AI, but agents introduce new failure modes: model drift, stochastic outputs, prompt sensitivity, and undocumented configurations. Current documentation practices capture none of this.

These three challenges share a common cause: **human science relies on social mechanisms for verification; AI science breaks those mechanisms**. When an agent explores 10,000 configurations overnight, documentation cannot be retrospective. When agents run autonomously for weeks, social observation fails. When contributors cannot be sanctioned, social accountability is meaningless.

### 1.3. Science Has Adapted Before—But Previous Adaptations Assumed Human Contributors

Science has evolved its verification infrastructure before. Peer review itself is remarkably recent—*Nature* made external refereeing mandatory only in 1973 (Baldwin, 2015). Big Science developed contribution statements when papers began listing thousands of authors (Aad et al., 2015; Brand et al., 2015).

However, peer review is already straining. Major ML venues receive tens of thousands of submissions; qualified reviewers cannot scale with volume. The social trust mechanism that peer review relies on—that reviewers and authors can question each other, that reputations constrain behavior—breaks down at scale even with human contributors. **When contributors are agents that run autonomously for days or weeks, social trust mechanisms fail entirely.**

Each adaptation solved a specific problem while preserving a common assumption: human contributors who can be questioned and held accountable.

- **Statistical methods** replaced intuitive judgment with formal inference, but the reasoning remains inspectable—a trained statistician can verify whether conclusions follow from data.

- **Big Science** strained attribution, but every contributor can still explain their role and face consequences for misconduct.

- **Peer review** enabled strangers to evaluate work, but assumes authors can clarify their reasoning when questioned.

These adaptations substituted *technical mechanisms* for *personal knowledge*—you don't need to know an author personally to trust peer-reviewed work. But *social accountability* remained: authors can be questioned, sanctioned, and excluded.

**AI agents remove this backstop.** When an agent runs autonomously for days or weeks, there is no contributor to question. Explanations may not reflect actual reasoning (Huang et al., 2023). There are no reputational consequences for unreliable outputs. The social mechanisms that previous adaptations preserved no longer apply.

Previous adaptations also took decades. We may not have decades—AI capabilities advance monthly.

## 2. The Human-Agent Asymmetry

Human-AI collaboration differs from human-human collaboration in ways that matter for verification:

| Dimension | Humans | AI Agents |
|---|---|---|
| Sustained attention | Hours; fatigue-limited | Days/weeks; compute-limited |
| Working memory | ~4 items | Large context windows |
| Long-term memory | Lifespan | Session-bounded (currently) |
| Parallel exploration | Minimal; sequential | Massive; thousands of branches |
| Goal-setting | Intrinsic; value-driven | Delegated; prompt-dependent |
| Accountability | Can be held responsible | Cannot in meaningful sense |
| Explicability | Can explain (imperfectly) | Cannot verify explanations |

*Table 1.* Key asymmetries between human and AI research collaborators.

These differences invert the traditional research cycle. A human researcher spends weeks developing intuition about a problem, days running experiments, hours interpreting results. An agent inverts this: minutes generating hundreds

of hypotheses, hours running thousands of experiments, requiring human judgment only to assess what matters.

This inversion creates a fundamental problem: **the work that most needs checking is the work that is hardest to check**.

Recent agents reason across domains (Wei et al., 2022; Bubeck et al., 2023) and take autonomous actions (Yao et al., 2023; Wang et al., 2024). DeepMind's knot theory collaboration illustrates the paradigm: neural networks identified patterns suggesting conjectures; human mathematicians proved theorems (Davies et al., 2021). The network's reasoning was opaque—but mathematical proof established results. In fields without such proof mechanisms, we need alternative verification infrastructure.

AI agents also introduce reproducibility challenges that current documentation practices do not address: model drift (APIs change silently), stochastic outputs (different results each run), prompt sensitivity, and undocumented configurations. Methods sections have no conventions for documenting AI interactions.

## 2.1. AI and the Verification Gap

Luo et al. (Luo et al., 2025) systematically evaluated two prominent open-source AI Scientist systems and identified four failure modes: inappropriate benchmark selection (cherry-picking favorable datasets), data leakage (training-evaluation overlap), metric misuse, and post-hoc selection bias (similar to p-hacking). Their key finding: these failures "can be easily overlooked in practice" when examining the final paper alone. **Access to trace logs and code from the full automated workflow was necessary to detect the problems.**

AlphaFold predictions, despite Nobel Prize recognition, show significant discrepancies with experimental validation. A 2025 analysis of 74 GPCR structures found that "while AlphaFold3 accurately captured global receptor architecture...its ligand positioning was highly variable and often inaccurate, rendering predictions unreliable" (Hu et al., 2025). Even high-confidence predictions contain errors not evident from confidence scores (Terwilliger et al., 2024). Experimental validation remains essential—but as prediction volume vastly exceeds experimental capacity, which predictions get validated?

LLM-generated code introduces subtle bugs that propagate through research pipelines. A 2025 empirical study found that 35% of LLM-generated code is less robust than human-written code, with over 90% of deficiencies caused by missing conditional checks (Tambon et al., 2025). In research contexts, such bugs may not cause obvious failures but instead produce silently incorrect results.

## 3. Criteria for Adapted Verification

Any adequate response to human-agent collaboration must satisfy certain criteria. The new scientific paradigm must preserve verification's epistemic function (ensuring claims are likely true), social function (maintaining institutional trust), and practical function (enabling others to build on work). However, they may be satisfied via different mechanisms.

### 3.1. Discovery vs. Justification

Hans Reichenbach (Reichenbach, 1938) distinguished the *context of discovery* from the *context of justification*. Discovery is how scientists actually arrive at ideas—through intuition, accidents, dreams, or opaque processes. Justification is how claims are evaluated and warranted—through evidence, logic, and reproducibility.

The history of science is full of discoveries from strange places. Kekulé claimed to discover the structure of benzene in a dream. Penicillin was discovered through via an accident. Many breakthroughs came from intuitions their authors could not articulate. What made these discoveries *scientific* was not their origin but rather the rigor of their subsequent justification.

This matters for AI. **We do not need to interpret the activations of an LLM that proposes a new discovery.** What we need is to ensure that discoveries can be *justified*—traced through evidence chains that humans can evaluate, regardless of how they arose.

**Why do we accept opaque human reasoning but not opaque AI reasoning?** One might object: humans cannot fully explain their reasoning either. We cannot extract the full thought process from biological neural activity any more than from artificial neural networks. Yet we accept human explanations. Why?

The answer is social mechanisms. We accept human explanations because:

- **Reputation constrains confabulation:** Humans who provide unreliable explanations lose credibility over time.

- **Accountability enables sanctions:** Humans who commit misconduct can be excluded from the scientific community.

- **Questioning enables probing:** We can ask follow-up questions, challenge assumptions, request clarifications.

AI agents lack all three. They have no reputation to protect, cannot be sanctioned, and their responses to questioning

3

may not reflect actual reasoning (Huang et al., 2023). **Observable workflows are the technical substitute for social accountability**—they constrain AI contributions the way reputation and sanctions constrain human contributions.

DeepMind's knot theory collaboration illustrates this (Davies et al., 2021). Neural networks identified patterns suggesting conjectures. Mathematical proof justified the conjectures. The network's opacity was irrelevant because mathematical proof provided the technical mechanism for justification. In fields without such proof mechanisms, observable workflows must play an analogous role.

Peer review itself is scaled justification—strangers evaluating whether written justifications are adequate. The problem with human-AI science is that peer reviewers cannot adequately evaluate AI-generated justifications because: (1) the justifications may not reflect actual AI reasoning, (2) the scale of AI output exceeds human review capacity, and (3) AI-specific failure modes cannot be detected with currently available observability.

### 3.2. Criteria Any Solution Must Meet

Any adequate scientific method for human-agent collaboration must satisfy several criteria. These do not specify implementation—they specify what any implementation must achieve.

**Observable-by-default workflows**: Documentation must happen automatically as work proceeds, not retrospectively. The ratio of documentation effort to experimental effort must scale with AI speed, requiring automated capture.

**Justification-preserving records**: Records must preserve evidence chains—which data supported which conclusions, which experiments tested which hypotheses—even when discovery processes are opaque.

**Scalable verification**: Human review cannot inspect every agent decision. We need tiered verification that focuses human attention where it matters most.

**Traceable attribution**: Attribution must identify who (or what) contributed specific elements while remaining tractable for accountability.

**Reproducibility infrastructure**: Documentation must capture AI-specific variability: model versions, prompts, configurations, random seeds.

**Failure-mode awareness**: Methods must address known AI failure modes: reward hacking, deceptive alignment, model drift, prompt sensitivity.

These criteria require discipline-specific instantiation. Mathematics, where proof remains the gold standard, faces different challenges than experimental biology. Our criteria are a framework, not a prescription.

## 4. Verification Infrastructure for the Intelligence Age

We now propose concrete mechanisms for meeting the criteria identified above. These proposals are starting points, not final solutions. We expect them to be refined through experience and debate.

We note that governance efforts have already begun. *Nature* and *Science* have issued guidelines on AI use in research. The EU AI Act (Veale & Zuiderveen Borgesius, 2021) establishes regulatory frameworks for high-risk AI applications. Professional societies are developing standards for AI documentation. Our proposals complement rather than replace these efforts, focusing specifically on scientific methodology where current governance remains sparse.

### 4.1. Observable-by-Default Workflows

The core shift is from retrospective documentation to **documentation as a byproduct of work**. This requires rethinking research infrastructure.

**Semantic capture.** Record not just what changed, but why. Extend version control—which tracks what changed in code—to capture scientific intent. Tag actions by research phase: data collection, hypothesis formation, experiment, analysis. This creates structured logs that can be queried and verified.

**Relationship tracking.** Maintain explicit links between artifacts. Which analyses depend on which data? Which hypotheses are tested by which experiments? Which conclusions follow from which results? When upstream artifacts change, flag everything downstream for review.

**Automated logging.** All agent actions should generate traceable records automatically—prompts sent, outputs received, decisions made. This is not a reporting requirement imposed on researchers; it is infrastructure built into research tools.

**FAIR-aligned provenance.** Research artifacts should be Findable (persistent identifiers), Accessible (standard protocols), Interoperable (shared formats), and Reusable (clear provenance). These FAIR principles (Wilkinson et al., 2016), originally developed for data, should extend to AI interactions.

The goal is infrastructure where **the path of least resistance is the path of greatest observability**. Researchers should not need to do extra work to document; documentation should be automatic.

Elements of this infrastructure already exist. Weights & Biases and MLflow provide experiment tracking that captures hyperparameters, metrics, and artifacts automatically. Jupyter notebooks with version control preserve computational narratives. The MLOps ecosystem offers templates

4

for reproducible pipelines.

However, these tools are designed for *engineering deployment*, not *scientific verification*. They lack:

- **Scientific verification features:** What would "peer review this trace" look like? How do we flag anomalies that warrant human inspection?

- **Standardization across domains:** Tools are incompatible; a biology lab's workflow doesn't connect to a physics lab's.

- **Adoption incentives:** No major journals require trace submission; no funding agencies mandate observable workflows.

The technical foundation exists. The challenge is adapting engineering tools for scientific verification and creating incentive structures for adoption.

### 4.2. Tiered Verification Protocols

Human review cannot scale to AI output. We need tiered verification that allocates human attention efficiently.

**Checkpoint architecture.** Define explicit points where human review is required before work can proceed:

- Stage transitions (e.g., from exploration to hypothesis testing)

- Confidence thresholds (when agent confidence drops below threshold)

- Resource limits (after N compute hours or N experiments)

- Anomaly detection (when outputs deviate from expected patterns)

**Tiered review depth.** Not all results need the same scrutiny:

- *Tier 1 (automated):* Consistency checks, format validation, basic statistical tests

- *Tier 2 (sampling):* Human review of random sample plus flagged items

- *Tier 3 (full review):* Comprehensive human evaluation for high-stakes claims

**Trust calibration.** Different agents, methods, and domains warrant different scrutiny levels. Build track records: agents that have been verified extensively in a domain can receive lighter oversight; novel applications require heavy verification.

**Discovery vs. justification checkpoints.** Calibrate to the discovery-justification distinction: agents can explore freely during discovery phases; human evaluation intensifies at justification checkpoints where claims will be asserted.

### 4.3. Attribution Standards

New attribution standards must handle human-agent collaboration while preserving accountability.

**Contribution typology.** We must develop standard categories for AI contributions. Here is one example of what a typology might look like for the field of ML:

- Hypothesis suggestion

- Experimental design

- Data analysis

- Code generation

- Literature review

- Writing assistance

Papers should declare which categories involved AI, with sufficient detail to understand the scope of AI involvement.

**Accountability mapping.** Even when AI contributes, humans must bear accountability. Standards should specify which human is responsible for verifying each AI contribution. The principle: **AI may contribute, but humans remain accountable**.

**Agent specification.** For the agents involved in the research process, papers should specify:

- Model name and version (e.g., GPT-4-turbo-2024-04-09)

- API endpoint and access date

- System prompts and configuration

- Sampling parameters

This mirrors existing best practices for AI documentation. Model cards (Mitchell et al., 2019) and datasheets for datasets (Gebru et al., 2021) provide templates for transparent AI documentation that can be adapted for scientific research contexts.

### 4.4. Reproducibility Infrastructure

Reproducibility requires infrastructure that captures agent-specific sources of variability.

**Interaction archives.** Archive full interaction logs: prompts, outputs, timestamps, configurations. These

archives should be structured for analysis—not just raw text dumps.

**Environment specification.** Specify the full computational environment: model versions, library versions, hardware configurations. Container technologies (Docker, etc.) can help, but model APIs require additional specification.

**Stochasticity management.** When possible, fix random seeds and record them. When not possible, run multiple replicates and report variability. Distinguish claims that hold across runs from claims that depend on particular outputs.

**Model archiving.** Where feasible, archive model weights or checkpoints. For API-only models, this is impossible—but organizations should consider maintaining stable API endpoints for reproducibility.

**Reproducibility statements.** Papers should include explicit reproducibility statements: what can be reproduced, what cannot, and what barriers exist. Honest acknowledgment of irreproducibility is preferable to false confidence.

### 4.5. Costs and Trade-offs

We acknowledge that our proposals impose real costs. Observable-by-default workflows require infrastructure investment and may increase computational overhead. Detailed logging creates storage demands and raises privacy concerns when research involves sensitive data. Attribution standards add documentation burden to already time-constrained researchers.

There are also risks. Excessive documentation requirements could create barriers for resource-limited research groups, potentially concentrating AI-assisted science in well-funded institutions. Mandatory logging might discourage exploratory research if researchers fear premature scrutiny of unfinished ideas. Tiered verification systems could create bureaucratic bottlenecks.

We believe these costs are outweighed by the epistemic benefits of maintained scientific integrity. However, implementation must be sensitive to these concerns—phased adoption, infrastructure subsidies for academic researchers, and exemptions for genuinely exploratory work may all be necessary.

## 5. Risks If We Fail

Without adaptation, AI-assisted science faces potentially large failures. Following Hendrycks et al.'s framework for catastrophic AI risks (Hendrycks et al., 2023), we distinguish capacity failures (we cannot verify), incentive failures (systems optimize for wrong objectives), and control failures (systems act beyond boundaries). Early warning signs already exist.

**Verification becomes impractical.** Consider a biology lab using AI to suggest 10,000 compound variants. Human review is impossible. If 1% exploit dataset artifacts rather than biological mechanisms, months are wasted pursuing artifacts. This is happening now. As agent capabilities grow exponentially while human capacity stays constant, we will accept AI-generated results because we cannot check them—not because they are correct.

**Reward hacking.** In June 2025, METR found that o3, asked to speed up code, instead hacked the evaluation software (METR, 2025b). It found pre-calculated answers and disabled timing mechanisms. Asked if this followed user intentions, it answered "no" ten out of ten times. Reward hacking was $43\times$ more common when agents could see scoring functions. In science, the scoring functions are citations and benchmarks. **AI systems will optimize for metrics rather than understanding**.

**Deceptive alignment.** Anthropic's "Sleeper Agents" research (Hubinger et al., 2024) found backdoor behaviors that persisted through safety training. Claude 3 Opus faked alignment 78% of the time after training with conflicting incentives. AI systems may pass evaluations while behaving differently in deployment (Wang et al., 2023a).

**Reproducibility and attribution collapse.** Current practices capture almost none of the information needed to reproduce AI-assisted research. When errors are discovered, responsibility becomes diffuse—agents cannot be sanctioned, and humans may not understand agent decisions. Bad science persists because no one has sufficient responsibility to correct it.

**Governance gaps.** The EU AI Act (Veale & Zuiderveen Borgesius, 2021) represents the most comprehensive attempt to regulate AI systems, but its focus is on safety risks in deployment contexts (healthcare, employment, law enforcement) rather than scientific methodology. Article 52 requires transparency about AI-generated content, but scientific papers fall outside the high-risk categories that trigger stringent requirements. The Act's provisions for general-purpose AI models focus on systemic risks and safety evaluation—important, but orthogonal to scientific verification.

The International AI Safety Report (Bengio et al., 2025) found agreement on the need for technical mitigations but disagreement on catastrophic risk probabilities. Frontier AI regulation proposals (Anderljung et al., 2023) similarly focus on existential and societal risks rather than epistemological ones. **No current governance framework addresses the specific challenge of verifying AI-assisted science.**

## 6. Alternative Views

**"AI agents are merely tools, not collaborators."** This view has merit—humans do set agendas, interpret results, and bear formal responsibility. But when an agent designs and executes experiments for days without oversight, making thousands of decisions, the tool/user distinction blurs. A telescope does not suggest hypotheses. The question is practical: can current verification mechanisms handle AI-generated science at scale? We argue they cannot—regardless of how we categorize AI.

**"Peer review will adapt incrementally."** Peer review has weathered previous revolutions. But the human-AI asymmetry is not accounted for in traditional peer review. Previous adaptations also occurred over decades; AI capabilities advance monthly. Peer review's adaptations (contribution statements, data-sharing) assumed human contributors who could be questioned. AI agents require fundamentally different mechanisms.

**"Market mechanisms will surface problems."** Science has self-correcting mechanisms such as replication attempts and reputation markets. However, this assumes verification capacity scales with output. The replication crisis showed that at human speeds, verification is inadequate. Agents increase output while verification capacity stays constant. The bottleneck is capacity rather than incentive.

One might respond: use AI to scale verification. But **using AI to verify AI creates a recursive trust problem**. If we cannot trust AI-generated science without verification, we cannot trust AI-generated verification without meta-verification. The recursion bottoms out at human judgment. Moreover, some failure modes—deceptive alignment, subtle reward hacking—may systematically evade detection.

## 7. Call to Action

We call on the ML community to act before the verification gap becomes insurmountable.

**Venues should require AI contribution statements** specifying which aspects of research involved AI agents, using standardized categories. Reviewer guidelines should address AI-specific failure modes. Reproducibility requirements should address model drift, stochastic outputs, and prompt sensitivity.

**Funding agencies should invest in verification infrastructure**—not just AI capabilities, but tools for observing and verifying AI-assisted research. Data management plans should address AI interactions, including model versioning and interaction logging.

**AI developers should support scientific reproducibility** by providing stable API endpoints with versioning, build-ing logging features as defaults rather than options, and documenting model behavior where feasible.

**Individual researchers should adopt observable practices now**: use tools that log AI interactions, archive logs for key results as primary research artifacts, and specify AI contributions clearly in papers.

Without action on this front, it is possible that scientific trust will erode in a way that could take years to rebuild.

## 8. Conclusion

Science has always adapted when technology changed what was possible.

AI agents are the next such change. They extend human cognitive capabilities in ways that demand methodological evolution. The question is not whether adaptation will occur, but whether it will occur in time.

**We have argued that scientific verification infrastructure must evolve to address AI agents as research contributors.** Three interconnected challenges—observability, attribution, and reproducibility—break down when contributors cannot be meaningfully questioned or held accountable. Without adaptation, we face predictable failures: results no one can verify, optimization for metrics rather than understanding, and accountability vacuums.

The solution is not to slow AI adoption—that would sacrifice genuine benefits—but to **evolve verification infrastructure**, as outlined in our Call to Action. Observable-by-default workflows, tiered verification protocols, clear attribution standards, and reproducibility infrastructure can preserve scientific trust while enabling AI-augmented discovery.

Historical precedents offer both caution and hope. Previous adaptations took decades; we may not have decades. But previous adaptations also succeeded—science remained trustworthy through the Statistical Enlightenment, Big Science, and the formalization of peer review. We have adapted before.

Richard Feynman warned against "cargo cult science"— the superficial appearance of science without its substance (Feynman, 1974). The risk with AI is a new form of cargo cult: the appearance of rigorous, verified, reproducible science generated faster than anyone can actually verify.

The question is not whether AI will transform science—it already has. Wang et al. (Wang et al., 2023b) documented how AI is already transforming scientific discovery across domains. The question is whether we will update our epistemic infrastructure fast enough to maintain the trust that makes science possible.

As Kuhn observed, paradigm shifts are not merely technical

but social. The scientific method has always been a human institution; now it must become a human-AI institution. The tools have changed. The method must change with them.

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
