# OpenReview forum: "Position: The Age of AI Agents Demands A New Scientific Paradigm To Sustain Trustworthy Science"
_ICML.cc/2026/Position_Paper_Track — ICML 2026 Position Paper Track regular_

### Official Review · Reviewer_QeYt · 2026-03-07

**Significance:** 4
**Argument Clarity:** 3
**Rating:** 5
**Confidence:** 4

**Questions:**

The authors propose verification infrastructures as a future mechanism to reduce review burdens, yet this overlooks the immediacy of the current crisis. Recalling Section 1.1, the verification gap is already acute—expert reviewers for top ML conferences are both scarce and time-constrained, often unable to assess papers with the care required. What measures do you think can help narrow down this gap?

**Alternative Views Section:**

Yes

**Compliance With Llm Reviewing Policy A Conservative:**

Affirmed.

**Discussion Potential:**

3

**Paper Summary:**

This paper points out an important issue we are now facing: The AI-generated content is beyond human review capabilities.  The authors point out three challenges for human-AI science  (can we see what happened? Who is responsible? Can we verify results?)

The authors give a vision for future verification infrastructure, including
- Observable agent workflow, and automatic documentation.
-  A tiered verification protocol is given.
- Clear contribution attribution
- Reproducibility infrastructure.

**Position:**

Yes

**Position In Title:**

Yes

**Related Work:**

3

**Strengths And Weaknesses:**

The paper presentation is clear. The challenges and problems are raised promptly.

**Support:**

3

---

> ### Author Rebuttal · Authors · 2026-03-31
>
> We thank you for the review.
>
> **Q. The paper proposes future mechanisms but overlooks the immediacy of the current crisis. What measures can help narrow the gap now?**
>
> The Call to Action (Section 7) already addresses this, but we will make the near-term vs. long-term distinction more explicit. Several measures require no new infrastructure and can happen in the next 1–2 review cycles:
>
> - *Structured AI contribution statements* at venues, going beyond current binary LLM policies to specify which research phases involved agents, which models were used, and what human oversight was applied. This requires only updated submission forms.
> - *Interaction log archiving* as supplementary material, similar to how code and data are currently shared. Researchers using agents for core contributions can archive prompts, outputs, and key decision points today with existing tools.
> - *Automated checks* for known AI-specific failure modes: post-hoc benchmark selection, data leakage in automated pipelines, sensitivity to agent reruns. As Luo et al. demonstrated, these checks are necessary and currently absent.
>
> Medium-term (1–3 years): standardized trace formats for cross-tool interoperability, and tiered verification pilots at workshops before scaling to major venues.
>
> We will sharpen this distinction in the revision.

---

> > ### Author Rebuttal · Reviewer_QeYt · 2026-04-07
> >
> > Let me rephrase my question. Given the tons of papers without traces/logs, how could AI help responsible human reviewers?

---

### Official Review · Reviewer_CxTJ · 2026-03-13

**Significance:** 3
**Argument Clarity:** 3
**Rating:** 5
**Confidence:** 4

**Questions:**

Please refer to weakness.

**Alternative Views Section:**

Yes

**Compliance With Llm Reviewing Policy A Conservative:**

Affirmed.

**Discussion Potential:**

3

**Paper Summary:**

This paper presents a position that science must evolve its verification infrastructure, as it has before with peer review. This is an important and timely and important topic. It proposes criteria for an adapted verification infrastructure that emphasizes observable by-default workflows, scalable verification, and clear attribution. A concrete mechanism is proposed for meeting these criteria. The mechanism has a good potential draw attention and discussion from the community.

**Position:**

Yes

**Position In Title:**

Yes

**Related Work:**

3

**Strengths And Weaknesses:**

Strength:

1. This paper studies an important and timely topic. Evolving the verification infrastructure of science is of emergent needs in the agent era. It worth attention from machine learning community, since the submissions keep growing.

2. The motivation is clear and it is supported by concrete evidence. This makes this paper solid.

3. The proposed criteria looks sound and the proposed mechanism looks reasonable enough to draw attention from the community.

Weakness:

The presentation of this paper needs improvement. This paper is full of short paragraphs and itemized contents. This makes the presentation looks fragmented and incomplete. Furthermore, table 1 violates the margin.

**Support:**

3

---

> ### Author Rebuttal · Authors · 2026-03-31
>
> We thank you for the review. We will fix table 1's formatting and reduce the number of bullet lists!

---

> > ### Author Rebuttal · Reviewer_CxTJ · 2026-04-01
> >
> > My concerns in the first round of review are minor.  After rebuttal, I am still positive about this paper.

---

### Official Review · Reviewer_ZXvi · 2026-03-15

**Significance:** 4
**Argument Clarity:** 3
**Rating:** 4
**Confidence:** 4

**Questions:**

1. Operational definition of the verification gap
The paper introduces the concept of the “verification gap.” Could the authors provide a more formal or operational definition of this concept, possibly including measurable indicators that could track its growth over time?

2. Relationship to existing reproducibility initiatives
How do the proposed infrastructure elements relate to existing efforts such as ML reproducibility checklists, experiment tracking platforms, or open science frameworks? Are the authors proposing extensions to these systems or entirely new infrastructure?

3. Adoption incentives
Many of the proposed practices (e.g., full interaction logs, detailed AI attribution) may increase the burden on researchers. What mechanisms or incentives might realistically encourage widespread adoption across the research community?

4. Use of AI for verification
The paper briefly mentions the recursive trust problem of using AI to verify AI-generated results. Could the authors elaborate on potential architectures for hybrid human–AI verification systems that mitigate this recursion problem?

**Alternative Views Section:**

Yes

**Compliance With Llm Reviewing Policy A Conservative:**

Affirmed.

**Discussion Potential:**

4

**Paper Summary:**

This paper presents a critical issue concerning the future of scientific practice in the era of autonomous AI research agents. The authors aim to examine a central concept: the widening “verification gap” between the scale of scientific outputs generated by AI systems and the human community’s capacity to verify them. The paper argues that as AI agents increasingly generate hypotheses, design experiments, write code, and produce research artifacts autonomously, traditional scientific verification mechanisms—such as peer review, reputation systems, and reproducibility practices—may no longer be sufficient.

The paper identifies three core challenges introduced by AI-assisted research: observability (whether the reasoning and actions of agents can be inspected), attribution (who is responsible for scientific contributions made by AI), and reproducibility (whether results generated through AI workflows can be independently verified). Building on these challenges, the authors propose a set of criteria for a new scientific verification infrastructure. These include observable-by-default workflows, scalable verification procedures, traceable attribution mechanisms, and improved reproducibility infrastructure tailored for AI-assisted research.

The paper also outlines several concrete proposals, including automated logging of AI interactions, tiered verification protocols combining automated and human review, standardized AI contribution statements, and archival practices for model configurations and interaction traces. The authors conclude with a call to action for the machine learning community, research venues, funding agencies, and AI developers to adapt scientific infrastructure before the verification gap grows too large.

**Position:**

Yes

**Position In Title:**

Yes

**Related Work:**

3

**Strengths And Weaknesses:**

Strengths

1.	Timely and important topic

The paper raises a highly relevant issue for the machine learning community: the implications of increasingly autonomous AI systems for the scientific process itself. As AI-assisted discovery and automated research pipelines become more common, questions around verification, reproducibility, and accountability are indeed becoming more pressing.

2.	Clear articulation of the “verification gap”

The paper provides a clear conceptual framing of the verification gap between AI-generated scientific outputs and human verification capacity. The framing is intuitive and supported by several motivating examples from contemporary AI systems and AI-assisted research tools.

3.	Structured conceptual framework

The organization around three challenges—observability, attribution, and reproducibility—provides a useful conceptual structure for thinking about human–AI scientific collaboration. This structure helps connect philosophical discussions of scientific methodology with practical ML infrastructure issues.

4.	Concrete proposals

Unlike many purely speculative position papers, the paper proposes actionable ideas such as:
- observable-by-default research workflows
- automated logging of agent interactions
- tiered verification protocols
- standardized AI contribution statements
- reproducibility infrastructure capturing model configurations and prompts

These suggestions make the paper more constructive and actionable for the ML community.

5. Inclusion of alternative views

The paper includes a dedicated section discussing alternative perspectives (e.g., AI as tools rather than collaborators, incremental adaptation of peer review), which is appropriate and strengthens the argument.

⸻

Weaknesses

1. Evidence is largely conceptual rather than empirical

While the paper presents a compelling narrative, most of the arguments rely on conceptual reasoning and selected anecdotes rather than systematic empirical evidence. For example, the magnitude of the projected verification gap is extrapolated from limited indicators and may be speculative.

2. Limited discussion of existing infrastructure efforts

Although the paper references tools such as experiment tracking systems and FAIR principles, the discussion could engage more deeply with existing reproducibility initiatives in ML and computational science. Some of the proposed mechanisms already partially exist in MLOps ecosystems, open science frameworks, or reproducibility programs.

3. Scope may be broader than necessary

The argument spans philosophy of science, AI governance, reproducibility infrastructure, and scientific sociology. While this breadth is intellectually interesting, it sometimes reduces the depth of analysis in specific areas (e.g., detailed technical mechanisms for scalable verification).

4. Practical feasibility remains unclear

Several proposals—such as universal logging of agent interactions or mandatory workflow observability—may introduce significant overhead or privacy concerns. The paper acknowledges some trade-offs but does not provide a detailed analysis of feasibility, incentives, or adoption pathways.

5. The notion of a “new scientific paradigm” may be overstated

While the paper raises important methodological challenges, it is not entirely clear whether these changes require a fundamentally new scientific paradigm or rather extensions of existing reproducibility and documentation practices.

**Support:**

3

---

> ### Author Rebuttal · Authors · 2026-03-31
>
> We thank the reviewer for providing a detailed review. We address each point below.
>
> **W1. Evidence is largely conceptual rather than empirical.**
>
> The paper already includes empirical case studies: Luo et al.'s systematic evaluation of AI Scientist failures, AlphaFold validation discrepancies against experimental GPCR structures, METR's data on agent capability doubling times, and Liu et al.'s study finding 35% of LLM-generated code is less robust than human-written code. These are published findings documenting verification failures that are already occurring.
>
> **W2. Limited discussion of existing infrastructure efforts.**
>
> We respectfully note that Section 4.1 already engages with existing infrastructure: we discuss Weights & Biases, MLflow, Jupyter notebooks with version control, the MLOps ecosystem, and FAIR principles (Wilkinson et al., 2016). We also explicitly identify the gap between these engineering-oriented tools and what scientific verification requires. Our proposals are framed as extensions of these systems, not replacements. What is genuinely novel is the discovery-justification checkpoint distinction, agent-specific failure mode awareness, and the argument that social accountability mechanisms require technical substitutes when contributors cannot be questioned or sanctioned.
>
> **W3. Scope may be broader than necessary.**
>
> We believe the breadth is necessary for the argument. The Reichenbach distinction (Section 3.1) is load-bearing: it establishes why opaque AI discovery is acceptable as long as justification remains rigorous, which directly motivates our criteria. Similarly, the governance discussion (Section 5) is needed to show that no existing framework addresses scientific verification specifically, which is the gap our proposals fill. Narrowing to only technical mechanisms would miss the central point that the challenge is sociotechnical.
>
> **W4. Practical feasibility remains unclear.**
>
> Several of our proposals require no new infrastructure at all. AI contribution statements need only updated submission forms. Interaction log archiving works with existing tools (W&B, MLflow). Extended reproducibility checklists add a few fields. Section 4.5 already discusses costs and trade-offs, including phased adoption and exemptions for exploratory work. We will clarify the near-term vs. long-term distinction more explicitly, but we note that venue requirements have historically been effective forcing functions: data availability statements became standard within a few years of being required.
>
> **W5. "New scientific paradigm" may be overstated.**
>
> We use "paradigm" deliberately. Every previous adaptation of scientific verification (statistical methods, Big Science contribution statements, peer review) preserved social accountability as a backstop: contributors could be questioned, sanctioned, and excluded. AI agents remove this backstop entirely. The shift from social to technical mechanisms for verification is, we argue, qualitatively different from extending existing practices. Individual components may be extensions, but the need for them to *substitute* for social accountability rather than complement it is what makes this a paradigm-level change.
>
> **Q1. Operational definition of the verification gap.**
>
> This is a good point. We will add this formalization: the ratio of scientific claims produced per unit time to claims that can be meaningfully verified per unit time. Measurable proxies include submission-to-qualified-reviewer ratio, fraction of papers with reproducible results, time-to-first-replication, and proportion of AI-assisted claims undergoing post-publication verification. We will add this to Section 1.1.
>
> **Q2. Relationship to existing reproducibility initiatives.**
>
> See W2 above. Our proposals extend existing systems to address agent-specific challenges. The key additions are documentation of prompts, model versions, and interaction logs; the discovery-justification checkpoint framework; and automated-by-default rather than retrospective documentation.
>
> **Q3. Adoption incentives.**
>
> See W4 above.
>
> **Q4. Hybrid human-AI verification architectures.**
>
> AI can handle Tier 1 verification (consistency checks, statistical validation) where correctness has objective ground truth that humans can independently confirm, e.g., "does the reported p-value match the data?" The AI verifier's output is checkable even if the original science is not. Tiers 2 and 3 retain human judgment for subjective assessments, though may be assisted by AI assuming the human gives final confirmation.
>
> We are happy to discuss any of these points further.

---

> > ### Author Rebuttal · Reviewer_ZXvi · 2026-04-04
> >
> > Thank you for the detailed and thoughtful rebuttal. I find that several of my key concerns have been addressed in a constructive way.
> >
> > In particular, the clarification of the operational definition of the “verification gap” is helpful and significantly strengthens the paper. The proposed formulation and associated measurable proxies make the concept more concrete and empirically actionable.
> >
> > I also appreciate the clarification that the proposed infrastructure is intended as an extension of existing reproducibility and MLOps systems, rather than a replacement. The distinction between engineering tooling and scientific verification requirements is now clearer, and the discovery–justification checkpoint framing is a useful addition.
> >
> > The discussion of feasibility and adoption, especially the point that many elements can be layered onto existing tools and that venue requirements can serve as incentives, improves the practical plausibility of the proposal.
> >
> > That said, I still have a few follow-up questions and points that would benefit from further clarification:
> >
> > 1. Empirical grounding of the verification gap
> >
> > While the rebuttal points to several relevant empirical studies, the connection between these examples and the magnitude and growth rate of the verification gap remains somewhat indirect. Could the authors further clarify how these pieces of evidence support the projected scaling behavior of the gap, rather than illustrating isolated failure modes?
> >
> > 2. Relationship to existing infrastructure (more concretely)
> >
> > The rebuttal clarifies that the proposal extends existing systems, but it would be helpful to see a more explicit mapping. For example, which components of current tools (e.g., experiment tracking, reproducibility checklists) correspond to each proposed criterion, and where exactly are the missing pieces?
> >
> > 3. Adoption incentives and burden distribution
> >
> > While venue requirements are a plausible mechanism, there may be uneven costs across different research settings (e.g., resource-constrained groups vs. large labs). Could the authors elaborate on how adoption might be incentivized without disproportionately increasing barriers to entry?
> >
> > 4. Hybrid human–AI verification and recursion limits
> >
> > The tiered verification framework is clearer now, but the recursive trust issue remains subtle. Could the authors further clarify under what conditions AI-based Tier 1 verification can be reliably trusted, and how failure modes in the verifier itself would be detected?
> >
> > Overall, I find the paper stronger after the rebuttal and maintain a positive assessment, but these points would benefit from further clarification in the final version.

---

### Decision · Program_Chairs · 2026-04-30

**Decision:**

Accept (regular)

**Comment:**

This paper received three positive reviews (1 borderline accept, 2 accept). There is general appreciation for the clarity and timeliness of the position the paper takes, clarity of exposition -- in particular of the emerging "verification gap", and several concrete proposals for addressing the gap. There were a few concerns raised during the review process, most of which were addressed by the authors' responses. Given these, an accept consensus was reached.